# Lung Adenocarcinoma Presenting as a Soft Tissue Metastasis to the Shoulder: A Case Report

**DOI:** 10.3390/medicina57020181

**Published:** 2021-02-20

**Authors:** Kazuhiko Hashimoto, Shunji Nishimura, Masao Akagi

**Affiliations:** Department of Orthopedic Surgery, Kindai University Hospital, 377-2 Ohno-Higashi, Osaka-Sayama City, Osaka 589-8511, Japan; shunnisi@med.kindai.ac.jp (S.N.); makagi@med.kindai.ac.jp (M.A.)

**Keywords:** lung cancer, soft tissue metastasis, lung adenocarcinoma

## Abstract

*Background:* Metastasis to soft tissue is rare, and the pathogenesis remains unclear. Soft tissue metastases (STMs) have varied presentations; existing reports are few. Herein, we report a case of STMs of the shoulder with a rich characterization. *Case presentation:* A 93-year-old man presented to our hospital with pain and swelling of the left shoulder for one week. Magnetic resonance imaging (MRI) showed a T1 low-intensity and T2 high-intensity mass. We suspected a primary sarcoma and performed a needle biopsy. However, on histopathological examination, the findings were suggestive of lung adenocarcinoma. Fluorine-18 fluorodeoxyglucose (FDG) positron emission tomography-computed tomography also revealed FDG accumulation in the right lung, thus confirming the diagnosis. *Conclusion:* Oncologists should keep in mind that STMs of lung cancer may resemble soft-tissue sarcomas at the time of initial diagnosis.

## 1. Introduction

Metastasis to soft tissue is rare, and its pathogenesis is unclear [1]. Metastasis to muscle was first reported in 1854 AD [2]. According to previous studies, metastasis to soft tissue mostly occurs in lung cancer, followed by kidney and colon cancers [3,4].

Soft tissue metastases are indicative of terminal-stage cancers [3,5]. Soft tissue metastases are diagnosed by autopsy in 0.75–9% of cancer specimens [6,7,8].

Soft tissue metastasis originating from lung cancer is rare, with a reported overall prevalence of 2.3% [9]. Soft tissue metastases of lung cancer can be found in the back muscles, chest, abdomen, thighs, and paraspinal muscles [6,9,10].

Herein, we report the case of an elderly patient who developed lung cancer metastasis to the soft tissue of the left shoulder, which was diagnosed before the primary lesion with pathological images and positron emission tomography-computed tomography (PET-CT) findings.

## 2. Case Presentation

A 93-year-old man noticed a swelling and pain in his left shoulder one week prior to his consultation at our hospital. He came because of a rapid increase in the pain and size of the mass. The size of the mass at presentation was approximately 10 cm × 8 cm. The skin overlying the mass was red and tender. The pain prevented movement of his left shoulder; the abduction angle was only 20°. Magnetic resonance imaging (MRI) showed a low intensity on T1-weighted images and high intensity on T2-weighted images (Figure 1A,B). We suspected a primary soft tissue sarcoma and therefore performed a needle biopsy of the mass. On histological examination, findings suggestive of adenocarcinoma were observed upon staining the biopsy specimen with hematoxylin and eosin (Figure 1C). On immunohistological staining, cancer cells that were positive for p53, CAN5/2, and CK7 were observed (Figure 1D–F).

Afluorine-18 fluorodeoxyglucose PET/CT (18F-FDG-PET/CT) was performed. It revealed a mass in the left shoulder with 18F-FDG accumulation and a maximum standardized uptake value of 18.4 (Figure 2A). In addition, 18F-FDG accumulation in the right lung was observed (Figure 2B). Based on the pathology and F-18 FDG-PET/CT results, we diagnosed soft tissue metastasis of lung adenocarcinoma in the left shoulder. Radiotherapy (8 Gy) was administered for the soft tissue metastatic lesions. Written consent was obtained from the patient for the publication of this case report and the supporting images.

## 3. Discussion

The liver, brain, bone, and kidney are the most frequent sites of lung cancer metastasis [3,4]. To the best of our knowledge, only one case of lung cancer metastasis to the shoulder ([1,9,11,12]; Table 1) has been reported. This is the first detailed case report of a large soft tissue metastasis of lung cancer to the shoulder as the initial manifestation of the disease.

The mechanisms of metastasis to soft tissue like muscles include hematogenous and lymphatic spread [9,13]. Trauma may be another mechanism, as suggested in a previous report [14]. Our patient had no history of trauma; therefore, the possible mechanism of metastasis was through the left deltoid branch of the subclavian artery or left suprascapular artery.

Soft tissue metastases are painful in 83% of cases and palpable in 78% [12]. In addition, soft tissue metastasis is the initial manifestation of cancer in 27% of cases [1]. In general, if the tumor size is ˃5 cm, there is a high possibility of malignancy [15]. The sizes of soft tissue metastases are often <5 cm, and occasionally, their surface may be ulcerative or exudative [1,16]. Shoulder metastasis of lung cancer can initially present as a small nodule (approximately 1.5 cm) ([1]; Table 1).

Our patient had a relatively large mass, which was painful and palpable, but not ulcerative. It is important to differentiate an unknown primary cancer from primary malignancies, such as soft tissue sarcomas, if the patient presents with only soft tissue metastases at the time of initial diagnosis, like in our patient.

On MRI, soft tissue metastases present as low-signal images on T1-weighted sequences and high-signal images on T2-weighted sequences, with poorly defined margins [17,18]. Soft tissue sarcomas have varied presentations. They often invade the bones and surrounding tissues of the extremities with unclear margins on MRI [19]. Therefore, it is difficult to determine whether a patient has a primary soft tissue sarcoma or a metastatic soft tissue tumor based on MRI findings alone, like in our patient.

Histopathological examination is useful for examining the primary site of the cancer and determining the specific histological type [20,21,22]. CAM5.2, CK7, and TTF are as useful markers for lung adenocarcinoma [23]. In the current case, we suspected that the primary tumor might be an adenocarcinoma of the lung after pathological examination of the biopsy specimen.

18-F-FDG-PET/CT is useful for detecting soft tissue metastasis and primary cancer lesions [24,25]. In a prior meta-analysis, the detection sensitivity of 18-F-FDG-PET/CT for localizing the primary lesion was 84–92%, and the specificity was high (71–84%); therefore, it is useful for identifying lesions that cannot be detected by CT or MRI [24,25,26,27]. We made a definitive diagnosis using 18-F-FDG-PET/CT and the histopathological findings. The management of such cases is generally determined by the primary tumor, disease stage, and physical state of the patient [9]. Chemotherapy and radiotherapy are the basic treatments for soft tissue metastasis, and surgery is used for only selected cases [3,9]. Radiotherapy is effective in relieving pain in soft tissue metastasis [11]. We administered radiotherapy to our patient to relieve local pain because the patient was quite old and unable to tolerate chemotherapy [28].

Previous studies have reported that the survival of patients with soft tissue metastases ranges between months and three years after diagnosis [9]. Consequently, careful follow-up is necessary.

## 4. Conclusions

The oncologist should keep in mind that lung adenocarcinoma may present as a soft tissue sarcoma at the time of initial diagnosis.

## Figures and Tables

**Figure 1 medicina-57-00181-f001:**
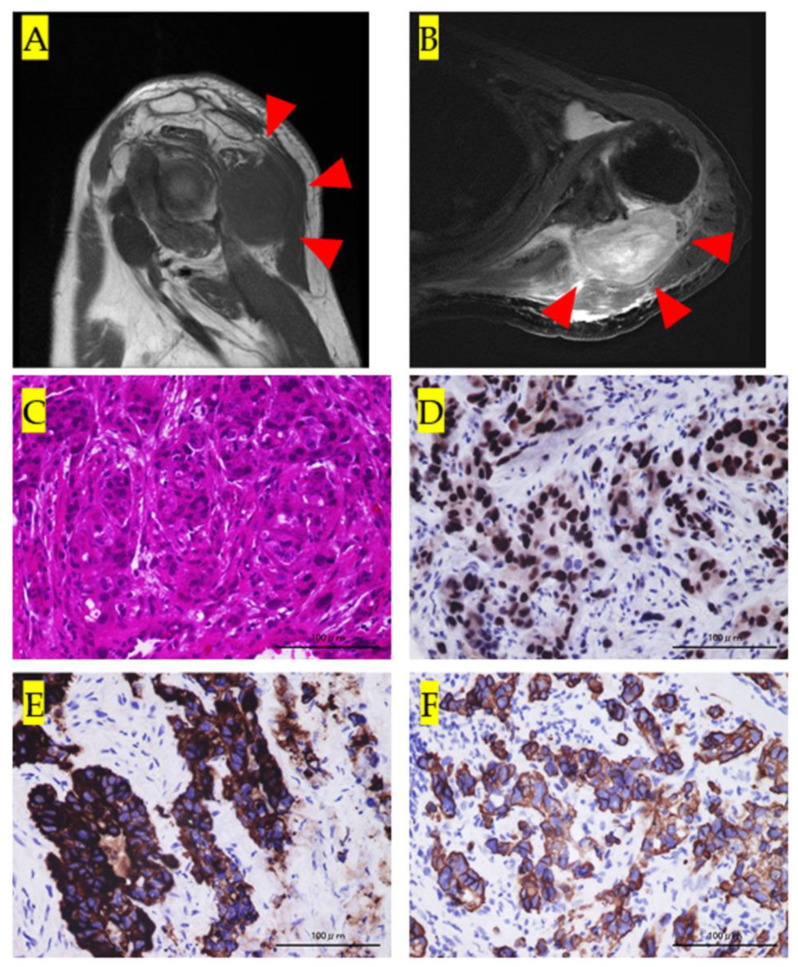
(**A**): Sagittal view of the left shoulder on T1 magnetic resonance imaging (MRI) showing a low-intensity tumor mass (red arrowheads). (**B**): Axial view of the left shoulder on T2 MRI showing a high-intensity tumor mass (red arrowheads). (**C**): Histological findings upon staining with hematoxylin and eosin. The histology shows highly heterozygous cancer cells with a luminal structure. (**D**): Immunohistological staining for p53. Cancer cells positive for p53 can be observed in the specimen. (**E**): Immunohistological staining for CAM5.2. Cancer cells positive for CAM5.2 can be observed in the specimen. (**F**): Immunohistological staining for cytokeratin (CK) 7. Cancer cells positive for CK7 can be observed in the specimen.

**Figure 2 medicina-57-00181-f002:**
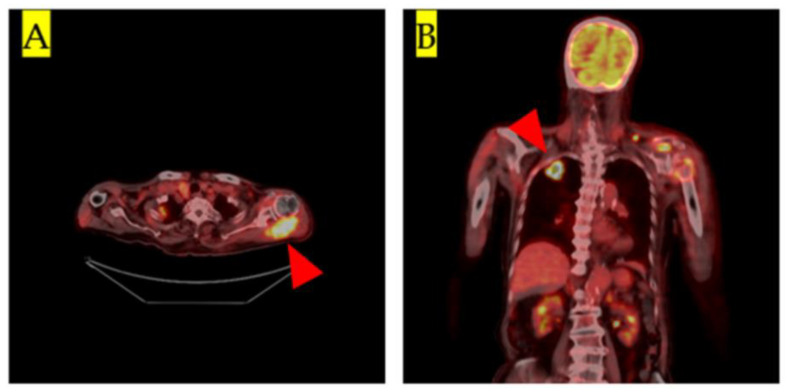
(**A**): Coronal view of fluorine-18 fluorodeoxyglucose positron emission tomography/computed tomography (18F-FDG-PET/CT) showing accumulation of 18F-FDG in the tumor mass in the left shoulder (red arrowhead). The maximum standardized uptake value was 18.4. (**B**): Frontal 18F-FDG-PET/CT. Accumulation of 18F-FDG in the right lung tumor lesion is visible (red arrowhead).

**Table 1 medicina-57-00181-t001:** Clinical features of soft tissue metastasis of lung cancer.

Year/Author	Age (Years)	Size (cm)	Pain	Deep or Subcutaneous	Diagnostic Imaging Tools	Histology	Metastasis at Initial Presentation
2007/Plaza JA, et al.	56	1.7 × 1.6 × 1.5	N/A	N/A	N/A	Adenocarcinoma	Yes
Current case	93	10 × 8 × 7	Yes	Deep	MRI, PET-CT	Adenocarcinoma	Yes

N/A: not applicable, PET-CT: positron emission tomography-computed tomography, MRI: magnetic resonance imaging.

## Data Availability

The datasets used and/or analyzed during the current study are available from the corresponding author on reasonable request.

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
