# Peer review of "Lung Adenocarcinoma Presenting as a Soft Tissue Metastasis to the Shoulder: A Case Report"

_medicina, 2021, doi:10.3390/medicina57020181_

Round 1
Reviewer 1 Report
The proposed manuscript is a case report of lung cancer soft tissue metastases ; this is a quite rare clinical condition but there are numerous articles that report this pathological situation.
The case report is well organized with good iconography and sufficient references.
Manuscript that reports a rare clinical condition but already reported in numerous other manuscripts. I believe that, given the rich iconography, it deserves to be learned as a case report.
Author Response
Author’s response:
Thank you for reviewing our paper. We have revised it according to your insightful comments. We believe that the paper has tremendously benefitted from your insights and has become more informative. We have also improved the English language of the paper. We hope you will review it again.
Reviewer #1
Comments and Suggestions for Authors
The proposed manuscript is a case report of lung cancer soft tissue metastases ; this is a quite rare clinical condition but there are numerous articles that report this pathological situation.
The case report is well organized with good iconography and sufficient references.
Manuscript that reports a rare clinical condition but already reported in numerous other manuscripts. I believe that, given the rich iconography, it deserves to be learned as a case report.
Author’s response:
Thank you for pointing this out.
As you pointed out, there are similar reports, but I believe this is the first detailed case report that includes pathological and PET-CT imaging findings. Therefore, we think this point needs to be emphasized.
Author’s action:
We have revised the sentence in the abstract part and the introduction part as follows:
In the abstract part:
Herein, we report a case of STMs of the shoulder with rich iconography. (Line:11-12)
In the introduction part:
Herein, we report the case of an older patient who developed lung cancer metastasis to the soft tissue of the shoulder, which was discovered before the primary lesion was diagnosed with detailed information, including pathological images and PET-CT imaging findings.
(Line:33-36)
Reviewer 2 Report
The authors presented a case of 93-year-old patient who developed lung cancer metastasis to the soft tissue of the shoulder which was discovered before the primary lesion was diagnosed.
Very well presented case with excellent Figures. Although similar case has been published previously it is worth of reading. Case presentation is fluent and concise followed by well structured discussion. References are up to date. Quality of English is excellent (above average).
I do not have any major remarks, just one minor objection:
The patient had adenocarcinoma of the right lung, and metastasis was in left shoulder. In discussion the authors suspected the possible mechanism of metastasis through the deltoid branch of the subclavian artery or suprascapular artery. Please clarify which artery left or right.
Author Response
Author’s response:
Thank you for reviewing our paper. We have revised it according to your insightful comments. We believe that the paper has tremendously benefitted from your insights and has become more informative. We have also improved the English language of the paper. We hope you will review it again.
Reviewer#2
Comments and Suggestions for Authors
The authors presented a case of 93-year-old patient who developed lung cancer metastasis to the soft tissue of the shoulder which was discovered before the primary lesion was diagnosed.
Very well presented case with excellent Figures. Although similar case has been published previously it is worth of reading. Case presentation is fluent and concise followed by well structured discussion. References are up to date. Quality of English is excellent (above average).
I do not have any major remarks, just one minor objection:
The patient had adenocarcinoma of the right lung, and metastasis was in left shoulder. In discussion the authors suspected the possible mechanism of metastasis through the deltoid branch of the subclavian artery or suprascapular artery. Please clarify which artery left or right.
Author‘s response:
Thank you for pointing this out.
As you pointed out, it should be stated if it is left or right.
We believe that the tumor in the lung went into the bloodstream of the whole body, returned to the heart, and formed metastases through the left deltoid branch of the subclavian artery or left suprascapular artery on the left.
Author’s action:
We have revised the sentence in the discussion part as follow:
In the current case, the patient had no history of trauma; therefore, the possible mechanism of metastasis was through the left deltoid branch of the subclavian artery or left suprascapular artery. (Lines 81-83)
Reviewer 3 Report
Line 24 ( first line of introduction needs to be reformatted).
line 25. Replaced avoid provided with reported and add AD after he had 1854.
Line 24 and 25 needs revision.
Line 29 and 31. Arthur reports that soft tissue metastasis in lung cancer is extremely rare but given the prevalence of 2.3%. This needs further justification.
Line 31 and 32. Suggest replacing the word "preferred". Suggest commonly " reported sites".
The language and grammar needs extensive revision. Needs revision from native English speaker are language editing software.
This case is not unusual. In our experience, we see several soft tissue metastasis because of lung cancer.
Author Response
Author’s response:
Thank you for reviewing our paper. We have revised it according to your insightful comments. We believe that the paper has tremendously benefitted from your insights and has become more informative. We have also improved the English language of the paper. We hope you will review it again.
Reviewer #3
Comments and Suggestions for Authors
Line 24 ( first line of introduction needs to be reformatted).
line 25. Replaced avoid provided with reported and add AD after he had 1854.
Line 24 and 25 needs revision.
Author‘s response: Thank you for pointing out this. We should revise and reformat the sentences in Lines 24-25.
Author’s action:
We have revised the sentences in the Lines 24-25 as follows:
The first description of muscle metastasis was reported in 1854 AD [2]. According to previous studies, lung cancer is the most likely cancer to give rise to soft tissue metastasis following kidney and colon cancers [3,4]. (Lines 24-26).
Line 29 and 31. Arthur reports that soft tissue metastasis in lung cancer is extremely rare but given the prevalence of 2.3%. This needs further justification.
Author‘s response: Thank you for pointing out the contradiction. The logical inconsistency needs to be justified.
Author’s action:
We have revised the sentence in Lines 29-31 as follows.
Soft tissue metastasis originating from lung cancer is uncommon, with a reported overall prevalence of 2.3% [9]. (Line 30-31)
Line 31 and 32. Suggest replacing the word "preferred". Suggest commonly " reported sites".
Author‘s response: Thank you very much for pointing this out.
We agree that “preferred” should be corrected to “reported”.
Author’s action:
We have revised the sentence in Lines 31-32 as follow.
The reported sites of the soft tissue metastasis of lung cancer are the muscles of the back, chest, abdomen, thighs, and paraspinal column [6,9,10]. (Lines 31-32)
The language and grammar needs extensive revision. Needs revision from native English speaker are language editing software.
Author‘s response: We agree with the reviewer’s suggestion that the language and grammar should be extensively revised by a native English speaker.
Author’s action: We have revised and proofread the English grammar in this manuscript.
This case is not unusual. In our experience, we see several soft tissue metastasis because of lung cancer.
Author‘s response: Thank you for pointing this out.
As you pointed out, there are similar reports, but I believe this is the first detailed case report that includes pathological and PET-CT imaging findings. Therefore, we think this needs to be emphasized.
Author’s action:
We revised the sentence in the abstract part and in the introduction part as follow.
In the abstract part:
Herein, we report a case of STMs of the shoulder with rich iconography. (Line:11-12)
In the introduction part:
Herein, we report the case of an older patient who developed lung cancer metastasis to the soft tissue of the shoulder, which was discovered before the primary lesion was diagnosed with detailed information, including pathological images and PET-CT imaging findings.
(Line:33-36)
Round 2
Reviewer 3 Report
(Soft tissue metastasis of cancer is a rare neoplasm whose pathology remains unclear) needs revision. Suggest Metastasis to soft tissue is rare.
The manuscript still needs revision in language and grammar
Author Response
Thank you very much for reviewing our manuscript again. We have revised the manuscript according to your suggestion. We hope that our paper is more comprehensible, and we are open for more comments from you.
Reviewer#3
Comments and Suggestions for Authors
(Soft tissue metastasis of cancer is a rare neoplasm whose pathology remains unclear) needs revision. Suggest Metastasis to soft tissue is rare.
The manuscript still needs revision in language and grammar
Authors’ response and actions:
Thank you for pointing this out.
We have changed “Soft tissue metastasis of cancer is a rare neoplasm whose pathology remains unclear” to “Metastasis to soft tissue is rare, and its pathogenesis is unclear.” (Line25-26)
Also, we submitted our manuscript to a professional medical editing service for a thorough editing to improve the language and grammar. We hope that it has been greatly improved as required.